# Transoral Robotic Surgery for Oral Cancer: Evaluating Surgical Outcomes in the Presence of Trismus

**DOI:** 10.3390/cancers16061111

**Published:** 2024-03-10

**Authors:** Ting-Shen Lin, Ci-Wen Luo, Tsai-Ling Hsieh, Frank Cheau-Feng Lin, Stella Chin-Shaw Tsai

**Affiliations:** 1Department of Medical Education, Tungs’ Taichung MetroHarbor Hospital, Taichung 43503, Taiwan; t14412@ms3.sltung.com.tw; 2Department of Medical Research, Tungs’ Taichung MetroHarbor Hospital, Taichung 43503, Taiwan; t14825@ms3.sltung.com.tw (C.-W.L.); t14159@ms3.sltung.com.tw (T.-L.H.); 3Department of Otolaryngology, Tungs’ Taichung MetroHarbor Hospital, Taichung 43503, Taiwan; 4School of Medicine, Chung Shan Medical University, Taichung 40201, Taiwan; 5Department of Surgery, Chung Shan Medical University Hospital, Taichung 40201, Taiwan; 6Superintendent Office, Tungs’ Taichung MetroHarbor Hospital, Taichung 43503, Taiwan; 7Department of Post-Baccalaureate Medicine, National Chung Hsing University, Taichung 40227, Taiwan

**Keywords:** trismus, oral cancer, transoral robotic surgery (TORS), overall survival, disease-free survival, concurrent chemoradiotherapy, radiotherapy, betel quid, propensity score matching

## Abstract

**Simple Summary:**

Trismus is a common complication in oral cancer patients that may influence surgical outcomes. This retrospective matched cohort analysis examined the influence of preoperative trismus on survival outcomes in 40 male Taiwanese oral cancer patients undergoing transoral robotic surgery (TORS). Through 1:1 propensity score matching, 20 trismus patients were compared to 20 non-trismus controls. TORS demonstrated comparable short-term surgical outcomes in trismus patients compared to non-trismus patients. There were no significant differences between groups in operation time, blood loss, margin status, flap reconstruction rates, duration of nasogastric tube feeding, or length of hospital stay. Analysis of long-term oncological outcomes demonstrated similar 5-year disease-free survival between groups, indicating comparable tumor control. However, trismus patients experienced significantly poorer 5-year overall survival. After adjusting for confounders, trismus independently conferred a 13-fold higher long-term mortality risk. Ultimately, this study highlights preoperative trismus as a harbinger of non-inferior short-term surgical outcomes, but diminished long-term survival in oral cancer patients treated robotically, even among those with equivalent tumor persistence following surgery. Further research should explore mechanisms linking trismus to mortality and approaches to improve outcomes in this population.

**Abstract:**

Trismus, defined as restricted mouth opening, is a common complication among Taiwanese oral cancer patients, especially those who chew betel quid. However, the impact of trismus on survival outcomes in oral cancer patients undergoing transoral robotic surgery (TORS) is unclear. This study aimed to investigate the associations between trismus and surgical outcomes in Taiwanese male oral cancer patients treated with TORS. We conducted a retrospective propensity score-matched cohort study of 40 Taiwanese male oral cancer patients who underwent TORS between 2016 and 2022. Overall, 20 patients with trismus were matched to 20 patients without trismus. TORS achieved similar operative and short-term clinical outcomes in trismus patients to non-trismus patients. There were no significant differences between groups in operation time, blood loss, margin status, flap reconstruction rates, duration of nasogastric tube feeding, or length of hospital stay. Kaplan–Meier and Cox proportional hazard regression analyses were performed to compare overall survival (OS) and disease-free survival (DFS) between the two groups. The overall survival (OS) rate at three years was significantly lower in patients with trismus than those without trismus (27.1% vs. 95.0%, log-rank *p* = 0.02). However, there was no significant difference in disease-free survival (DFS) rates between the trismus and non-trismus groups (36.6% vs. 62.7%, log-rank *p* = 0.87). After adjusting for confounders, trismus was independently associated with a 13-fold increased risk of mortality (adjusted HR 12.87, 95% CI 1.55–106.50, *p* < 0.05). In conclusion, trismus appears to be an independent prognostic factor for reduced long-term OS in Taiwanese male oral cancer patients undergoing TORS, though short-term surgical outcomes were non-inferior in the trismus patients. Further research is warranted to clarify the mechanisms linking trismus and survival in this population.

## 1. Introduction

Trismus, commonly characterized by restricted mouth opening, is frequently observed in patients with head and neck cancers [1,2,3,4,5]. In Taiwan, this condition is particularly prevalent among those who habitually chew betel quid, a practice known to contribute to oral submucosal fibrosis (OSMF) and oral cancer risk [6,7]. While the debilitating effects of post-treatment trismus on patients’ quality of life are well documented [8,9], the impact of preoperative trismus on surgical outcomes in oral cancer patients, particularly those undergoing transoral robotic surgery (TORS), remain underexplored. This oversight persists despite evidence suggesting that preoperative trismus, irrespective of the surgical method, poses significant challenges to surgical access [10,11,12,13,14]. The uncertainty surrounding how trismus influences the success of such therapies poses a significant challenge in managing oral cancer effectively.

Recent studies have provided insights into the broader implications of trismus. For instance, a 2021 study by Gondivkar et al. highlighted the correlation between trismus severity and reduced quality of life in oral cancer patients [9]. Another cross-sectional study emphasized the need for early identification and proactive management of trismus, thereby amplifying the potential for improved therapeutic outcomes [15]. Furthermore, a recent study by Chang et al. posited the promising role of TORS in enhancing treatment precision for oral cancer [16]. While magnification loupes can assist with visualization, TORS provides unparalleled optics and maneuverability via three-dimensional imaging and wristed instruments. These attributes help maximize tumor identification and negative margins [16,17], which are surgical features especially crucial within the limited oral cavity space in patients with trismus. For advanced tumors, TORS facilitates access to deep buccal and tongue base lesions considered inaccessible or overly morbid with traditional approaches [18].

Despite the existing research, there is a noticeable lack of studies elucidating the nuanced relationship between pre-treatment trismus and surgical outcomes in Taiwanese male oral cancer patients treated with TORS. This gap is significant, considering the prevalent use of betel quid in Taiwan related to non-oncological trismus and the rising adoption of surgical treatment alternatives [16,19]. Understanding this relationship is crucial for formulating comprehensive treatment strategies tailored to the unique challenges faced by this specific demographic and improving survival rates.

To address this research gap, our study utilized a retrospective analysis of medical records from male Taiwanese oral cancer patients who underwent TORS at a single teaching hospital between 2016 and 2022. Employing statistical methods to compare perioperative surgical outcomes and long-term disease-free and overall survival between those with and without trismus, our efforts to illuminate the influence of this common comorbidity on post-surgical outcomes will provide meaningful insight to inform clinical guidelines and decision-making.

## 2. Materials and Methods

### 2.1. Study Design and Participant Selection

In this retrospective cohort analysis, we examined patients with newly diagnosed, histologically confirmed oral cavity squamous cell carcinoma treated with curative-intent TORS by a single surgeon at a regional teaching hospital in Taiwan. The study period spanned from 1 January 2016 to 31 December 2022. Patients also had to have at least 12 months of follow-up data available for analysis of oncological outcomes. Exclusion criteria encompassed tumor involvement of the masticator space, unresectable or metastatic disease, involvement of bony structures such as the mandible or the maxilla, prior radiation therapy to the head and neck region, previous or synchronous malignancies within five years, previous significant head and neck surgeries, the upper or lower lip as the primary cancer site, and incomplete health records. From the eligible participants, a subset of 20 patients with trismus attributable to betel quid consumption were meticulously matched in a 1:1 ratio with control subjects devoid of trismus. Matching parameters included age, gender, anatomical location of the tumor, cancer stage, betel quid, alcohol, and tobacco usage. Trismus was characterized by varying degrees of maximum mouth opening (MMO), categorized as mild (30–35 mm), moderate (16–29 mm), and severe (≤15 mm) according to Thomas’s classification [20].

### 2.2. Data Collection

Data procurement involved a thorough review of electronic health records and the hospital’s cancer registry. Variables amassed for analysis comprised demographic details, clinical attributes, tumor specifics, operative details, adjuvant treatments, survival status, and functional outcomes. The primary outcome was overall survival (OS). Secondary outcomes were disease-free survival (DFS), surgical margins, postoperative function, and complications. The follow-up duration was set at five years until the occurrence of patient withdrawal, disease advancement, mortality, or the conclusion of the study period on 31 December 2022. Institutional review board approval was obtained for this study.

### 2.3. Therapeutic Interventions

The cohort underwent TORS utilizing the da Vinci Si/Xi^®^ robotic apparatus, all procedures being executed by a single surgeon with extensive expertise in the field. Figure 1 illustrates representative preoperative, intraoperative, and postoperative pictures depicting key aspects of our transoral robotic surgical techniques for patients presenting with trismus. The use of other treatment modalities, including concurrent chemoradiation therapy (CCRT), radiation, chemotherapy, immunotherapy, and targeted therapies, was meticulously recorded. These treatments were administered in the postoperative or 5-year study period for regional or metastatic recurrence of the disease.

### 2.4. Robot-Assisted Compartment Resection and Neck Dissection Technique

To achieve compartment resections in our cohort, the tumor specimen was resected via TORS. Complementarily to this, a horizontal cervical incision following the skin crease was made to complete the neck dissection with the robot. A key aspect was maneuvering the robot under the mandible bone. This created a continuum of the specimen from mouth to neck, enabling comprehensive treatment while preserving anatomy.

Similarly, the reconstructive team utilized this innovative route for free flap reconstruction, conducting the procedure via the oral route. Vascular anastomosis was performed by passing under the mandible bone to the key vascular structures in the neck, leveraging the same minimally invasive pathway.

In patients requiring neck dissections, a similar robot-assisted transcervical approach was employed, consistent with the minimally invasive surgical philosophy and facilitating an integrated strategy for both tumor resection and regional lymph node management.

### 2.5. Statistical Analysis and Methodology

For statistical evaluation, categorical variables were scrutinized using chi-squared tests, while continuous variables were assessed via two-tailed *t*-tests. Logistic regression was employed to discern factors correlating with the incidence of trismus. Survival outcomes, specifically DFS and OS, were contrasted between the cohorts utilizing Kaplan–Meier estimates and log-rank tests. Cox proportional hazard models were applied to evaluate the impact of trismus on mortality risk post-adjustment for potential confounders. Hazard ratios (HRs) and 95% confidence intervals (CIs) were computed. A *p*-value threshold of <0.05 was set for statistical significance. All analyses were conducted using SAS software, version 9.4 (SAS Institute, Cary, NC, USA).

## 3. Results

### 3.1. Sample Selection of the Study Participants

In this retrospective study, we evaluated 115 patients diagnosed with oral cancer who underwent transoral robotic surgery between 2016 and 2022. Thirty-five patients were excluded based on the exclusion criteria. Of the remaining 80 patients, 24 presented with trismus, while 56 did not, yielding an incidence of 30% for trismus. To ensure comparability between groups, a 1:1 matching technique was employed using propensity score matching [21], resulting in the selection of 20 patients from each group for further analysis (Figure 2).

### 3.2. Patient Demographic and Baseline Characteristics

The demographic and baseline characteristics of the patients are summarized in Table 1. Our study encompassed only male patients in both cohorts, each comprising 20 individuals. The average age was marginally lower in the trismus cohort (56.20 ± 14.11) compared to the non-trismus cohort (59.70 ± 10.60). However, this variance did not reach statistical significance (*p* = 0.19). There was no statistical difference in body mass index (BMI) between the two groups (*p* = 0.36). The trismus severity of the cohort was mild in no patients, moderate in 13 (65%) patients, and severe in 7 (35%) patients. The distribution of primary tumor locations was nearly identical between groups, with the most common sites being the buccal mucosa (50% trismus vs. 55% non-trismus) and tongue (30% vs. 25%).

Pathological T, N, and overall stages were also closely matched between the trismus and non-trismus cohorts. A majority of patients in both groups had stage IV disease (60% in each arm). While 40% of patients in each arm were diagnosed with the T4 classification in our study, the subjects predominantly reflected criteria other than bone invasion, such as tumor size, invasion into the facial skin, or involvement of deep/extrinsic muscles of the tongue. The groups had similar rates of tobacco (90% in both arms), alcohol (85% trismus vs. 90% non-trismus), and betel nut use (80% vs. 85%). The prevalence of major comorbidities such as diabetes, chronic kidney disease, and liver disease was also comparable between the two groups.

### 3.3. Surgical Outcomes between Oral Cancer Patients with and without Trismus after Undergoing Transoral Robotic Surgery (TORS)

In our investigation of factors associated with trismus in oral cancer patients undergoing TORS, various clinical parameters were examined (Table 2). Our logistic regression analyses encompassed tracheotomy rates, nasogastric tube feeding duration, surgical margin status, operation length, blood loss, length of hospital stay, and the need for reconstructive procedures and adjuvant therapies.

Tracheotomy was not performed in any patient with trismus, while 2 out of 20 patients without trismus (10%) underwent the procedure. The odds ratio for tracheotomy in patients with trismus when compared to those without approached zero, though with a notably wide confidence interval (0.001; 95% CI: 0.001–999.9), suggesting a rare but not statistically significant difference. Nasogastric tube feeding duration slightly varied, with trismus patients requiring an average of 12.50 days compared to 16.55 days for those without. This difference, represented by an odds ratio of 0.954 (95% CI: 0.891–1.022), was not statistically significant, indicating comparable postoperative recovery times in terms of feeding tube dependence.

Regarding surgical margin status, both groups showed a similar incidence of positive or close margins (5%), with an odds ratio of 1.000 (95% CI: 0.058–17.181). This parity demonstrated the consistent surgical precision achieved across both patient groups.

The length of the operation was slightly shorter for patients with trismus, averaging 237.70 min, versus 264.35 min for those without trismus. However, the minimal difference, with an odds ratio of 0.998 (95% CI: 0.993–1.003), did not reach statistical significance, suggesting that trismus does not considerably affect the total duration of the surgical procedures. Blood loss differed markedly between the two groups, with patients with trismus experiencing significantly less blood loss (43.75 mL) than those without trismus (210.5 mL). Despite the apparent difference, the odds ratio of 0.999 (95% CI: 0.996–1.002) suggests that this did not translate into a statistically significant variation, likely due to the high variability in blood loss among patients without trismus.

The hospital length of stay was marginally shorter for patients with trismus (19.35 days) than those without (21.80 days), with an odds ratio of 0.994 (95% CI: 0.963–1.026) indicating no significant difference between the two groups.

In terms of reconstructive procedures, the use of rotation flap, split-thickness skin graft, and free flap reconstruction showed no significant differences between patients with and without trismus, suggesting that trismus status does not necessarily dictate the complexity or type of reconstruction required post-TORS.

Lastly, the application of adjuvant therapies documented for the entire 5-year follow-up period, such as CCRT, radiotherapy, chemotherapy, cetuximab, and immune therapy (nivolumab/pembrolizumab), displayed no significant differences between the two groups. This indicates that trismus status does not affect the likelihood of requiring additional treatments post-surgery.

In sum, TORS demonstrated non-inferior outcomes in trismus patients to non-trismus patients. There were no significant differences between groups in operation time, blood loss, surgical margin status, flap reconstruction rates, tracheotomy use, duration of nasogastric tube feeding, or length of hospital stay. TORS was able to achieve similar operative and short-term clinical outcomes in trismus patients as in non-trismus patients. Importantly, TORS maintained key advantages as a minimally invasive, cosmetically superior approach for oral cavity tumors even in the setting of trismus, as there was no conversion to open surgeries. Likewise, the usage of adjuvant therapies, including concurrent chemoradiation (CCRT), radiotherapy, chemotherapy, cetuximab, and immunotherapy (nivolumab/pembrolizumab), was also equivalent between the two cohorts.

### 3.4. Factors Associated with the Survival of Oral Cancer Patients with and without Trismus Treated with TORS

Analysis of factors associated with survival outcomes are summarized in Table 3. On univariate Cox regression analysis, the presence of trismus was associated with significantly worse overall survival (HR 10.04, 95% CI 1.25–80.62, *p* < 0.05), but not disease-free survival (HR 1.09, 95% CI 0.37–3.27, *p* = 0.88). After adjusting for relevant clinical factors, including nasogastric tube duration, hospital stay, and use of concurrent chemoradiation therapy, trismus remained an independent predictor of decreased overall survival (adjusted HR 12.87, 95% CI 1.55–106.50, *p* < 0.05). Days of nasogastric tube feeding were not significantly associated with overall survival after multivariate adjustment (adjusted HR 1.09, 95% CI 0.98–1.21, *p* = 0.09). Other measured variables, including tracheotomy, surgical margin status, operation time, blood loss, hospital stay, flap reconstruction, adjuvant CCRT, radiotherapy, and the use cetuximab and nivolumab/pembrolizumab, were not independently predictive of disease-free or overall survival either. Notably, while concurrent chemoradiation and radiotherapy were given mainly in the postoperative period as adjuvant therapies, the use of cetuximab and nivolumab/pembrolizumab either sequentially or concurrently with chemotherapy was mostly for locoregional recurrence or distant metastasis occurring in the study period as a consensual decision made by the multidisciplinary team in our institution.

### 3.5. Disease-Free Survival (DFS) and Overall Survival (OS) between Oral Cancer Patients with and without Trismus Who Were Treated with Transoral Robotic Surgery (TORS)

The long-term survival outcomes of DFS and OS were assessed using Kaplan–Meier survival analysis and two-sided log-rank tests (Figure 3). The mean follow-up time for the investigated cohort was 34.3 months, ranging from 2 to 36 months. The three-year overall survival (OS) rate was markedly reduced in the cohort presenting with trismus relative to their non-trismus counterparts (27.1% vs. 95.0%, log-rank *p* = 0.02). Despite a discernible trend towards diminished disease-free survival (DFS) rates within the trismus group, statistical analysis did not reveal a significant disparity when contrasted with the non-trismus group (36.6% vs. 62.7%, log-rank *p* = 0.87). Observationally, disease-free survival (DFS) rates for patients with and without trismus demonstrate a close alignment during the initial 22 months, with a notable separation emerging after 24 months. This pattern suggests that for a substantial period, the disease-free interval between the groups remains indistinguishable. Subsequent statistical analysis, utilizing log-rank testing, corroborates this observation by revealing an absence of a significant difference in DFS rates between the trismus and non-trismus groups (*p* = 0.87). In contrast, the OS curve for patients with trismus showed a steeper decline when compared to their non-trismus counterparts. This trend suggested a more rapid decrease in survival rates for patients experiencing preoperative trismus. This observation was statistically reinforced through log-rank testing, which demonstrated that overall survival was significantly worse in patients with trismus (*p* = 0.02).

### 3.6. Pattern of Disease Recurrence

Our study meticulously examined the pattern of disease recurrence among patients, distinguishing between those with and without trismus at the time of diagnosis. In the cohort of patients presenting with trismus, a total of six individuals experienced disease recurrence post-treatment. Among these, there was a notable prevalence of regional metastasis, with five patients developing this form of recurrence. Only one patient in the trismus group encountered a local recurrence, suggesting that trismus may be associated with a higher likelihood of disease spreading beyond the primary site.

In contrast, the recurrence pattern in patients without trismus demonstrated a more balanced distribution. Out of the total recurrences in this group (*n* = 6), an equal number of patients (three each) developed local and regional recurrences. This indicates a different, perhaps less aggressive, pattern of disease progression in the absence of trismus.

## 4. Discussion

In this retrospective analysis of 40 patients undergoing transoral robotic surgery for oral cancer, our investigation sheds light on the nuanced interplay between preoperative trismus and surgical outcomes. Patients with and without trismus selected for this study demonstrated no significant differences in any measured demographic, clinical, pathological, or risk factor variable. The two cohorts were well matched, allowing analysis of trismus as an independent predictor of oncological outcomes. Notably, short-term surgical results were comparable among patients with and without trismus, emphasizing the procedural equipoise. However, a remarkable revelation emerged, as trismus, a challenging surgical consideration, was associated with significantly poorer 5-year overall survival despite achieving analogous tumor control. Rigorous adjustment for confounding variables revealed that trismus independently conferred a striking mortality risk, exceeding 10 times that of patients without trismus. Our findings indicated that trismus should be viewed not merely as a perioperative consideration but also as a critical prognostic indicator with profound implications for long-term survival in this population. As we delve into these outcomes, it becomes paramount to unravel the intricate mechanisms linking trismus to mortality and explore avenues for optimizing patient outcomes in this challenging clinical context. This study contributes valuable insights that merit consideration in the broader discourse on oral cancer management and warrant further investigation in the pursuit of enhanced therapeutic strategies.

In our study, the trismus patients all had a history of betel quid chewing, which is a well-established risk factor for OSMF and subsequent trismus in Taiwan and other regions where this habit is prevalent [6,7]. To isolate the effect of betel quid-related trismus, we excluded patients with tumors involving the masticator space, as well as those with prior radiation or major surgery that could contribute to trismus from other mechanisms. However, we did not quantify the duration or intensity of betel quid exposure, which may influence the severity of OSMF and the degree of trismus present preoperatively.

While the current study did not exhaustively evaluate trismus etiology, our findings raise important questions about how the causes of trismus intersect with oral carcinogenesis pathways and influence the biological behavior of these cancers. Prospective studies systematically collecting data on trismus risk factors, biomarkers, and their associations with survival are needed. Multidisciplinary efforts combining chronic disease epidemiology, molecular tumor analyses, and functional assessments may reveal key pathogenic mechanisms that could be therapeutically targeted to improve outcomes in this vulnerable population.

In addressing the inclusion of patients with stage IV disease, it is critical to delineate that while 60% of our study cohort was classified as having stage IV disease (Table 1), this predominantly reflected non-bone-invading aspects of T4a tumors. Our exclusion of patients with bone invasion and masticator space involvement was stringently observed. The rationale for this approach was to focus on the subset of T4 tumors that—despite their advanced classification—did not involve bone invasion, but met other criteria for stage IV classification, such as extensive local disease without bone involvement. This distinction is paramount for understanding the specific patient population under study and the surgical challenges and outcomes associated with TORS in this context. The approach allowed us to explore the outcomes of TORS in managing sizable or locally advanced oral cancers without the added complexity of bone involvement, which is beyond the scope of minimally invasive techniques like TORS.

A critical component of our surgical protocol for these selected advanced-stage patients, compartment resection was designed to ensure comprehensive tumor removal in a manner that respects the anatomical and functional integrity of the oral cavity and associated structures. In instances where neck dissection was indicated due to the extent of disease or nodal involvement, a separate cervical incision was strategically made. This incision was placed horizontally along a natural skin crease in the neck, thereby adhering to principles of cosmetic and functional preservation. A key aspect of our compartment resection technique was the ability to excise the tumor in continuity with the neck dissection specimen, employing the angled visual advantage of the robot and the flexibility of the EndoWrist, without necessitating a mandibulotomy. This approach allowed for the passage of the surgical specimen through the mandibular space, effectively maintaining the integrity of the mandible while ensuring a comprehensive oncological resection. This method not only facilitated a complete resection of the tumor and involved lymphatic tissue but also significantly reduced the procedure’s morbidity by avoiding the need for mandible reconstruction and the associated complications. Highlighting the efforts to balance oncological efficacy with the preservation of function and aesthetics, our approach reflects advancements in the surgical management of oral cavity cancers, aiming to improve patient outcomes through innovative and thoughtful surgical strategies.

Interestingly, as indicated in Table 2, the trismus cohort exhibited a similar duration of NG feeding and the need for tracheotomy compared to patients without trismus. The low incidence of tracheotomy in our cohort, despite treating advanced stage IV oral cavity disease, can be attributed to several factors. First, the precision and minimally invasive nature of TORS allows for less tissue disruption, reducing the need for tracheotomy as a preventive measure for airway management. Additionally, advancements in perioperative care and anesthetic techniques have enabled better management of the airway and breathing postoperatively, further decreasing the necessity for a tracheotomy [22]. In our series, tracheostomies were circumvented in the trismus cohort owing to meticulous airway handling intraoperatively during TORS and diligent postoperative monitoring, enabling early decannulation. Another contributing factor was the absence of any major bleeding episodes necessitating prolonged intubation. The criteria adopted for tracheostomy align with established protocols [23]. Despite the condition of preoperative trismus, the surgical procedure was able to retain the perioperative benefits of reduced morbidity and improved functional outcomes that characterize TORS.

Even though there were no massive intraoperative bleeding episodes in our cohorts, there was nearly five times more blood loss in the non-trismus group than the trismus group. Our observations regarding the differential blood loss between patients with and without trismus undergoing TORS for oral cancer may be further illuminated by considering the pathophysiological effects of betel quid chewing, a common habit in the populations studied. Betel quid chewing is strongly associated with the development of oral submucous fibrosis (OSMF), a condition characterized by progressive fibrosis of oral and paraoral tissue, leading to trismus and functional impairment [7,24,25]. The fibrotic process inherent to OSMF involves an excessive deposition of collagen fibers, resulting in a dense and avascular connective tissue layer. This fibrosis, while primarily noted for its clinical manifestation of restricted mouth opening, also implicates altered surgical outcomes, notably in terms of blood loss. The rationale for potentially reduced blood loss in trismus patients, as suggested by our findings, can be linked to the pathological changes induced by OSMF. Firstly, the reduced vascularity within fibrotic tissues, a direct consequence of the collagen overexpression and subsequent increase in connective tissue density, leads to fewer blood vessels being present in the surgical field [25]. Secondly, the stiffness and reduced elasticity of fibrotic tissues may limit the extent of trauma and consequently the bleeding during surgical manipulation. This backdrop of dense, fibrotic, and less vascular tissue in patients with trismus could logically contribute to the observed discrepancies in blood loss between our patient cohorts. Given these considerations, the marked fivefold difference in blood loss, despite not reaching statistical significance due to high variability, points to the complex interplay between preoperative conditions like trismus and the surgical landscape. It prompts a closer examination of fibrosis and its surgical implications, advocating for a tailored approach to manage oral cancer patients with a history of betel quid use. Further research focused on the surgical management of oral cancer in the context of OSMF and trismus is essential to validate these observations and refine surgical strategies accordingly.

Transitioning from the methodological rigor of our study to its clinical implications, it is pivotal to contextualize the observed DFS rates within the broader spectrum of TORS outcomes. The initial 22 months post-TORS exhibited no significant difference in DFS rates between patients with and without trismus, suggesting that TORS effectively manages the oncological challenges posed by trismus. The congruence in DFS during this period, despite the physical limitations imposed by trismus, represented TORS’s capability in achieving comparable margin statuses and oncological control. This alignment in early outcomes points towards the procedural advantages of TORS in navigating the anatomical constraints of trismus, reinforcing its role in maintaining effective cancer control. The absence of significant differences in long-term DFS rates further validates the efficacy of TORS in delivering equivalent oncological outcomes, irrespective of the presence of trismus. This observation emphasizes the importance of surgical innovation in enhancing treatment accessibility and effectiveness for patients facing complex clinical presentations. According to previous studies, some mechanisms may further explain the worse survival outcomes in oral cancer patients with trismus [9,26,27,28]. The potential impact of trismus on overall survival may be mediated through the emergence of dysphagia, culminating in malnutrition and the risk of aspiration [29]. These complications can exert a profound influence on a patient’s well-being, diminishing their quality of life and posing potential implications for long-term survival. Moreover, trismus, both as a preexisting condition and as a potential complication after surgery or subsequent adjuvant treatments, can complicate clinical follow-ups. This complexity introduces the risk of delays in diagnosing tumor recurrence or identifying a second primary cancer within the affected field, adversely affecting patient survival outcomes [2,30]. However, our study did not find a significant difference in the advanced tumor stage between the trismus and non-trismus groups: the disease-free survival between the two groups closely aligned, indicating no marked distinction.

Furthermore, delving deeper into the oral cancer recurrence pattern between the two groups, we uncovered that patients with trismus exhibited a predominance of regional metastasis, which might reflect a more aggressive or advanced disease state at diagnosis, potentially affecting prognosis and survival outcomes adversely. Conversely, the balanced recurrence pattern observed in the non-trismus group suggests a different trajectory of disease progression, with implications for patient management and follow-up strategies. While the sample was relatively small, these findings suggest that trismus may be associated with a distinct pattern of disease recurrence, potentially influencing overall survival outcomes. This insight provides a valuable foundation for future research aimed at unraveling the intricate relationship between trismus, recurrence patterns, and their impact on patient prognosis.

The utilization of radiotherapy in the treatment of oral cancer was not found to significantly influence survival in this cohort, contrary to some prior studies showing improved disease control with radiotherapy. Gomez et al. reported favorable outcomes with intensity-modulated radiotherapy as an adjuvant treatment post-surgery [31]. Another study reported that preoperative and postoperative radiotherapy resulted in higher disease-free survival rates than radiotherapy alone, suggesting the importance of combining modalities for optimal patient outcomes [32]. Furthermore, a recent study highlighted the effectiveness of definitive radiotherapy, either alone or with systemic therapy, for unresectable oral cancer [33]. These studies collectively demonstrated the crucial role of radiotherapy in improving DFS in oral cancer, mainly when used in conjunction with surgery or systemic therapy, and emphasized the necessity of appropriate dosing for maximized therapeutic benefit. The non-significant effect in our analysis may relate to the small sample and inherent biases of a retrospective study. The choice and dose of radiotherapy were not standardized between groups either. Further prospective investigation is warranted to clarify the impacts of adjuvant radiotherapy on survival after TORS, particularly in relation to trismus.

Our study’s extensive recording of treatments such as CCRT, radiation, chemotherapy, immunotherapy, and targeted therapies reflects a deliberate effort to understand their impact on overall survival. Given that these modalities were applied not just in the immediate aftermath of surgery but throughout the entire 5-year observation period, our analysis provides a comprehensive view of treatment effectiveness over the long term. This longitudinal approach allows us to compare survival outcomes between groups with a high degree of precision, taking into account the full spectrum of therapeutic interventions [34,35]. The inclusion of these treatments in our study highlights the importance of a multifaceted treatment strategy in improving survival outcomes for oral cancer patients, demonstrating our dedication to advancing the understanding of effective cancer care.

A considerable limitation of this study was the inability to standardize the surgical techniques used for addressing trismus intraoperatively. Although all patients were treated with TORS, the precise methods that the surgeon employed for releasing and expanding the oral aperture in the trismus group could not be controlled. Techniques included mucosectomy, myotomies or myomectomies of the masseter, masticator, and medial pterygoid muscle groups, the release of submucosal fibrosis, and other expansions at the surgeon’s discretion based on each patient’s presenting trismus severity [2,10,11,13,19,36,37]. The variability in approaches could have impacted postoperative outcomes. Future analyses may benefit from stratifying trismus patients by the specific intraoperative expansion techniques used and comparing survival rates and progression markers between these surgical method subgroups. Standardizing the trismus release procedures within a single study cohort could better isolate the effects of preexisting trismus itself on robotic surgery outcomes.

There are other limitations to this single-surgeon and single-institution study. First, the sample was small. Larger-scale analyses are needed to validate survival and treatment differences between trismus groups. Second, this was a retrospective study utilizing previously collected data. Prospective analyses can better control for confounders, such as precise treatment regimens between groups. Finally, functional outcomes like swallowing, oral hygiene, speech, and quality of life should have been addressed. Future studies should incorporate patient-reported measures to fully evaluate how trismus influences functional well-being after TORS.

## 5. Conclusions

This study found that preoperative trismus in oral cancer patients undergoing TORS was associated with comparable short-term surgical outcomes but significantly worse 5-year overall survival compared to matched patients without trismus. Trismus was an independent predictor of a mortality risk over tenfold, despite equivalent disease control between groups. The reasons for diminished survival remain unclear, but may relate to nutritional deficits, aspiration, and difficulties detecting recurrence. Further research should prospectively investigate functional outcomes and quality of life after TORS in trismus patients. Elucidating the mechanisms linking trismus to mortality and developing tailored interventions to optimize perioperative management may improve the prognosis of oral cancer patients with this common complication.

## Figures and Tables

**Figure 1 cancers-16-01111-f001:**
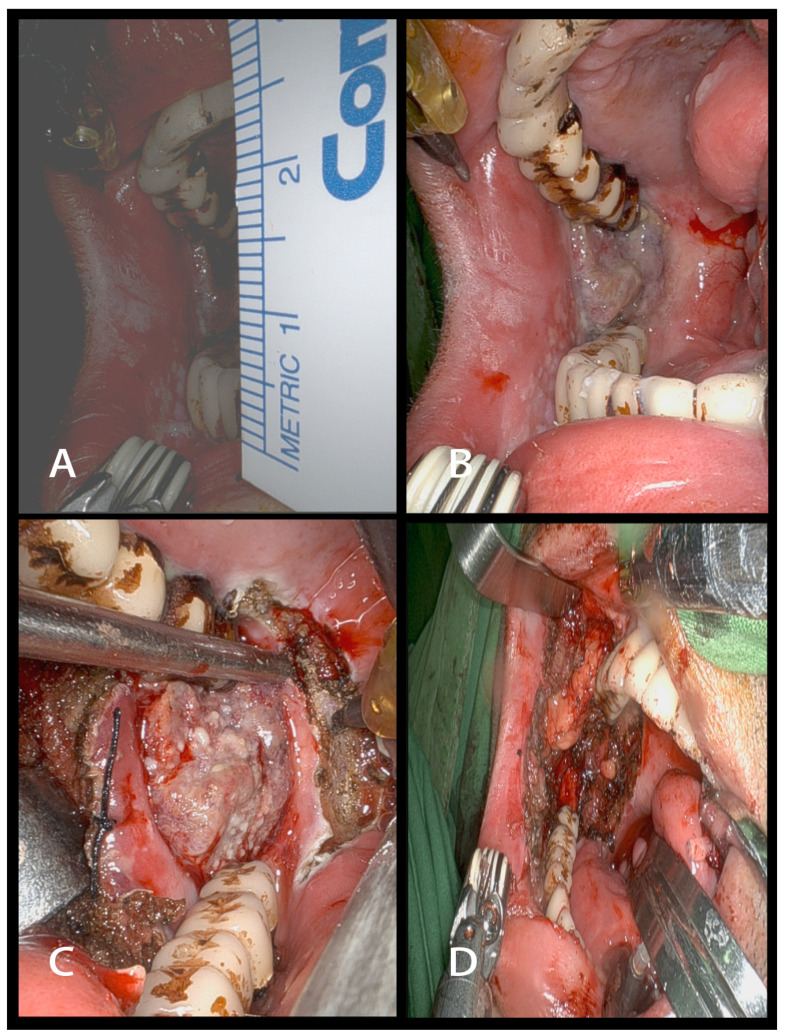
Surgical intervention in an oral cancer patient with trismus using transoral robotic surgery (TORS). (**A**) Preoperative image depicting a patient with moderate trismus due to oral cancer. (**B**) Intraoperative view showcasing surgical exposure during the excision of right buccal to retromolar trigone cancer. (**C**) Intraoperative image demonstrating tumor has been distinctly delineated. (**D**) The image depicts the appearance post-excision of buccal to retromolar trigone cancer.

**Figure 2 cancers-16-01111-f002:**
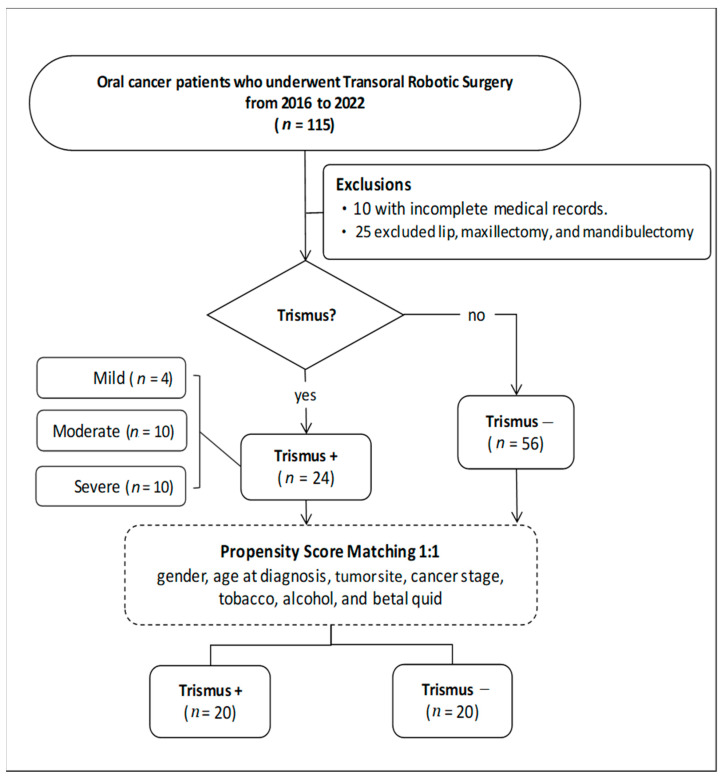
Flowchart of patient selection.

**Figure 3 cancers-16-01111-f003:**
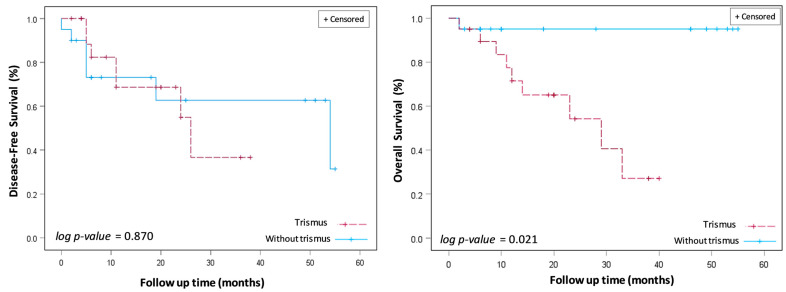
Disease-free survival (DFS) and overall survival (OS) between oral cancer patients with and without trismus who were treated with transoral robotic surgery (TORS).

**Table 1 cancers-16-01111-t001:** Baseline characteristics of oral cancer patients with/without trismus receiving TORS.

	Trismus		Without Trismus		*p*-Value
	(*n* = 20)		(*n* = 20)		
Gender					
Male	20	(100%)	20	(100%)	1.00
Age					
Mean ± SD	56.20 ± 14.11		59.70 ± 10.60		0.38
BMI					
Mean ± SD	25.96 ± 10.17		23.77 ± 3.09		0.36
Tumor location					
Buccal	10	(50%)	11	(55%)	0.94
Gingiva	1	(5%)	1	(5%)	
Mouth floor	1	(5%)	2	(10%)	
Retromolar	2	(10%)	1	(5%)	
Tongue	6	(30%)	5	(25%)	
Pathological T stage					
T1	3	(15%)	2	(10%)	0.96
T2	6	(30%)	7	(35%)	
T3	3	(15%)	3	(15%)	
T4	8	(40%)	8	(40%)	
Pathological N stage					
N0	6	(30%)	9	(45%)	0.32
N1	6	(30%)	3	(15%)	
N2	6	(30%)	7	(35%)	
N3	0	(0%)	1	(5%)	
NX	2	(10%)	0	(0%)	
Pathological staging					
I	2	(10.0%)	2	(10.0%)	1.00
II	3	(15.0%)	3	(15.0%)	
III	3	(15.0%)	3	(15.0%)	
IV	12	(60%)	12	(60%)	
Tobacco					
Yes	18	(90%)	18	(90%)	0.69
No	2	(10%)	2	(10%)	
Alcohol					
Yes	17	(85%)	18	(90%)	0.50
No	3	(15%)	2	(10%)	
Betel quid					
Yes	16	(80%)	17	(85%)	0.50
No	4	(20%)	3	(15%)	
Comorbidities					
Diabetes mellitus	6	(30%)	3	(15%)	0.26
Chronic kidney disease	3	(15%)	3	(15%)	1.00
Liver disease	3	(15%)	3	(15%)	1.00
Severity of Trismus				Numbers; Mean Range (mm)	
Mild				0; 0	
Moderate				13; 18.65	
Severe				7; 6.14	

**Table 2 cancers-16-01111-t002:** Logistic regression analyses of surgical outcomes in oral cancer patients with/without trismus treated with TORS.

	Trismus		Without Trismus		Odds Ratio (95% CI)
Tracheotomy					
Yes	0	(0%)	2	(10%)	0.00 (0.00–999.90)
No	20	(100%)	18	(90%)	Reference
Nasogastric tube feeding					
Day	12.50 ± 8.88		16.55 ± 9.99		0.95 (0.89–1.02)
Margin					
Positive/Close	1/0	(5%)	0/1	(5%)	1.00 (0.06–17.19)
Negative	19	(95%)	19	(95%)	Reference
Length of operation					
Minute	237.70 ± 105.77		264.35 ± 134.18		0.99 (0.99–1.00)
Blood loss					
mL	43.75 ± 72.13		210.50 ± 777.15		0.99 (0.99–1.00)
Hospital length of stay					
Day	19.35 ± 22.11		21.80 ± 18.73		0.99 (0.97–1.03)
Rotation flap					
Yes	1	(5%)	1	(5%)	1.00 (0.06–17.18)
No	19	(95%)	19	(95%)	Reference
STSG					
Yes	13	(65%)	12	(60%)	1.24 (0.34–4.46)
No	7	(35%)	8	(40%)	Reference
Free flap					
Yes	7	(35%)	7	(35%)	1.00 (0.27–3.67)
No	13	(65%)	13	(65%)	Reference
CCRT					
Yes	9	(45%)	11	(55%)	0.67 (0.19–2.33)
No	11	(55%)	9	(45%)	Reference
Radiotherapy					
Yes	11	(55%)	11	(55%)	1.00 (0.29–3.48)
No	9	(45%)	9	(45%)	Reference
Chemotherapy					
Yes	15	(75%)	18	(90%)	0.33 (0.06–1.97)
No	5	(25%)	2	(10%)	Reference
Cetuximab					
Yes	8	(40%)	9	(45%)	0.82 (0.23–2.86)
No	12	(60%)	11	(55%)	Reference
Nivolumab/pembrolizumab					
Yes	4	(20%)	4	(20%)	1.00 (0.21–4.71)
No	16	(80%)	16	(80%)	Reference

CI, confidence interval.

**Table 3 cancers-16-01111-t003:** Cox regression survival analysis of oral cancer patients with/without trismus treated with TORS.

	Disease-Free Survival		Overall Survival	
	Crude HR (95% CI)	Adjusted HR (95% CI)	Crude HR (95% CI)	Adjusted HR (95% CI)
Trismus (ref: non-trismus)	1.09 (0.37–3.27)		**10.04 (1.25–80.62)**	**12.9 (1.55–106.50)**
Tracheotomy	3.84 (0.47–31.42)		0.05 (0.00–999.9)	
NG days	1.02 (0.96–1.09)		1.08 (1.00–1.15)	1.09 (0.98–1.21)
Margin positive and close (ref: negative)	0.04 (0.00–398.40)		0.04 (0.00–811.30)	
Operation time	0.99 (0.99–1.00)		1.00 (0.99–1.00)	
Blood loss	1.00 (0.99–1.00)		1.00 (0.99–1.00)	
Hospital stay	1.01 (0.98–1.03)		1.02 (1.00–1.04)	1.00 (0.97–1.03)
Flap (ref: without)	0.89 (0.11–6.99)		0.96 (0.12–7.81)	
STSG (ref: without)	1.13 (0.37–3.39)		0.68 (0.19–2.36)	
Free-flap (ref: without)	0.45 (0.12–1.63)		1.03 (0.29–3.69)	
CCRT (ref: without)	2.50 (0.78–7.98)		2.15 (0.55–8.34)	
Radiotherapy (ref: without)	1.87 (0.58–5.97)		1.57 (0.40–6.08)	
Chemotherapy (ref: without)	1.45 (0.32–6.57)		29.27 (0.04–999.90)	
Cetuximab (ref: without)	2.19 (0.71–6.78)		2.05 (0.58–7.27)	
Nivolumab/pembrolizumab (ref: without)	0.42 (0.09–1.89)		1.05 (0.26–4.14)	
Adjustment for NG days, hospital day				
OR, odd ratio; CI, confidence interval				

## Data Availability

The data presented in this study are available at: https://doi_10.7910_DVN_2D5NMK (accessed on 26 January 2024).

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
