# Peer review of "Transoral Robotic Surgery for Oral Cancer: Evaluating Surgical Outcomes in the Presence of Trismus"

_cancers, 2024, doi:10.3390/cancers16061111_

Round 1

Reviewer 1 Report

Comments and Suggestions for Authors

Authors report the results of a retrospective propensity matching analysis investigating the surgical outcomes of transoral robotic surgery for oral cancers in the presence of trismus.

I commend the authors for an interesting topic of research. The results have been well-presented with appropriate conclusions.

One comment - Did the authors analyze the etiology of trismus? Would any of those factors played any role in the reduced survival in these patients?

Author Response

Reviewer 1:

Authors report the results of a retrospective propensity matching analysis investigating the surgical outcomes of transoral robotic surgery for oral cancers in the presence of trismus.

I commend the authors for an interesting topic of research. The results have been well-presented with appropriate conclusions.

One comment - Did the authors analyze the etiology of trismus? Would any of those factors play any role in the reduced survival in these patients?

Response:

Thank you for your positive comments and insightful review of our manuscript. We appreciate you highlighting this interesting topic and your feedback on the results and presentation.

Regarding your comment, we did not specifically analyze the etiology of trismus in this study. All patients with trismus had a history of betel quid chewing, which is a known risk factor for submucosal fibrosis and subsequent trismus development in our Taiwanese population. Tumors involving the masticator space were excluded to limit the etiology of trismus primarily to betel quid chewing-related submucosal fibrosis. While we matched the trismus and non-trismus groups for betel quid use, we did not collect detailed data on the duration or quantity of betel quid exposure that may have influenced trismus severity.

Additionally, we did not explore other potential contributors to trismus like prior radiation therapy or surgeries, as these were exclusion criteria for our study. Analyzing the specific etiologies and risk factors associated with trismus severity was beyond the scope of this initial work, but it is an excellent point that warrants further investigation. 

The observed reduction in long-term overall survival for trismus patients could potentially be influenced by the underlying causes and risk factor profiles that predisposed them to developing trismus in the first place. Evaluating these etiological factors and their interactions may shed more light on the mechanisms linking trismus to diminished survival. We agree that this line of inquiry would be valuable for future research efforts.

We have added two paragraphs in the Discussion:

“In our study, the trismus patients all had a history of betel quid chewing, which is a well-established risk factor for submucosal fibrosis and subsequent trismus in Taiwan and other regions where this habit is prevalent [6,7]. To isolate the effect of betel quid-related trismus, we excluded patients with tumors involving the masticator space, as well as those with prior radiation or major surgery that could contribute to trismus from other mechanisms. However, we did not quantify the duration or intensity of betel quid exposure, which may influence the severity of submucosal fibrosis and the degree of trismus present preoperatively.

While the current study did not exhaustively evaluate trismus etiology, our findings raise important questions about how the causes of trismus intersect with oral carcinogenesis pathways and influence the biological behavior of these cancers. Prospective studies systematically collecting data on trismus risk factors, biomarkers, and their associations with survival are needed. Multidisciplinary efforts combining chronic disease epidemiology, molecular tumor analyses, and functional assessments may reveal key pathogenic mechanisms that could be therapeutically targeted to improve outcomes in this vulnerable population.”

Thank you again for your thoughtful review. We appreciate you raising this important consideration regarding the role of trismus etiology, which will guide additional analyses as we continue exploring this topic.

Reviewer 2 Report

Comments and Suggestions for Authors

I read with interest this article on the impact of preoperative trismus in patients with oral cancer. The study aimed to investigate the associations between trismus and surgical outcomes in Taiwanese male oral cancer patients treated with TORS.

I have a number of observations:

·       The series of patients is modest. The same authors state that: “The non-significant effect in our analysis may relate to the small sample size and inherent biases of a retrospective study”.

·       Why did these patients have trismus? The authors' explanation on line 99 is "patients with trismus attributable to betel quid consumption" seems very hasty to me. Was the trismus due to fibrosis from betel quid or carcinoma caused by this habit?

·       On lines 54-57 the authors write: “While trismus is recognized for its debilitating effects on patients' quality of life [8,9], its impact on the surgical and survival outcomes of oral cancer patients, especially those undergoing advanced treatments like transoral robotic surgery (TORS), remains inadequately explored”.

This sentence seems ambiguous to me. If the authors intend to talk about preoperative trismus (as it should be, given the purpose of the study) the references cited (8 and 9) are inadequate, as both studies deal with the impact of postoperative trismus on the quality of life. If they mean post-treatment trismus, the topic is anything but “inadequately explored”. In addition to the two works cited (which analyzed 237 and 115 patients, respectively) many other studies have dealt with this topic. I don't think the type of surgical treatment (TORS or other techniques) matters. Preoperative trismus still creates access difficulties, whatever type of resection is used.

·       On lines 158-159 the authors write: “The majority of patients in both groups had stage IV disease (60% in each arm)”. The definition of a T4a carcinoma is: “”Tumor invades through the cortical bone of the mandible or maxillary sinus, or invades the skin of the face”. Why were 40% of the tumors analyzed T4? Furthermore, bone invasion was a reason for exclusion (line 95).. This contradiction needs to be clarified.

·       After tumor resection did the trismus disappear or not? It would seem not, given that on lines 250-254 the authors write: “Moreover, trismus has the potential to complicate clinical follow-ups, introducing the risk of delays in diagnosing tumor recurrence or identifying a second primary cancer in proximity to the affected field. This delay in diagnosis and subsequent treatment can adversely affect patient survival outcomes”. Furthermore, the sentence in lines 245-247 is ambiguous: “The potential impact of trismus on overall survival may be mediated through the emergence of dysphagia, culminating in malnutrition and the risk of aspiration”. It is not clear whether this sentence refers to pre- or post-operative trismus. These statements need to be clarified.

·       The secondary outcome was disease-free survival (DFS), but on the lines 185-186 and194 there is the “progression free survival” parameter. Are they two different parameters or the same one (in this case standardize the two terms)?

·       Each article presenting the results of the treatment of patients should include a sample sentence: “The mean follow-up time in the investigated cohort was X months, ranging from X to Y months”. In this article, however, I found only: “Patients also had to have at least 12 92 months of follow-up data available for analysis of oncologic outcomes” (lines 92-93). Moreover, I did not find the percentages of disease-free patients, those alive with disease and those who died from the disease or other causes. If I have to judge from figure 3, for patients with trismus it would seem that for both DFS and OS the follow-up stops at 40 months.

Furthermore, even for the DFS, patients with trismus fare worse than those without trismus, even if the difference, given the small number of patients, is not statistically significant. Furthermore, I wonder how it is possible that patients without trismus have almost 100% OS and much lower DFS. Would there have been many alive with disease? The lack of this data makes it difficult to interpret the results.

·       On line 193 and table 3 there is the sentence “cetuximab, and immunotherapy”. Since cetuximab is a monoclonal antibody, what do the authors mean by "immune therapy"?

·       On the lines 223-225 there is the sentence: “Rigorous adjustment for confounding variables revealed that trismus independently conferred a striking mortality risk, exceeding 10 times that of patients without trismus”, and on the lines 227-229 there is the sentence: “Although surgical margins and progression were equivalent between matched groups, trismus independently conferred over 10 times greater mortality risk”. It seems like a repetition of the same concept to me

·       The entire long paragraph on the indications for adjuvant CCRT (lines 256-284) seems absolutely useless and inconsistent with the purpose of the study.  I think that every head and neck surgeon should know the pros and cons of postoperative CCRT. All the guidelines give precise indications, also reporting the advantages and side effects.

·       On lines 318-319 there is the statement: “Additional work is also needed to clarify optimal adjuvant therapies in this setting”. As mentioned above, postoperative CCRT has precise indications in patients at risk (advanced pathological stage of the tumor, involved or close margins, N+, etc.). I don't think the fact that the tumor was resected with TORS or any other way should affect that treatment.

Comments on the Quality of English Language

Extensive editing of English language i required.

Author Response

I read with interest this article on the impact of preoperative trismus in patients with oral cancer. The study aimed to investigate the associations between trismus and surgical outcomes in Taiwanese male oral cancer patients treated with TORS.

I have a number of observations:

  • Comment 1: The series of patients is modest. The same authors state that: “The non-significant effect in our analysis may relate to the small sample size and inherent biases of a retrospective study”.

      Response 1: **Sample Size and Study Design**: We acknowledge the limitation regarding the modest sample size and the retrospective nature of our study. Future research with a larger cohort and prospective design is necessary to validate our findings and reduce biases inherent in retrospective analyses.

  • Comment 2: Why did these patients have trismus? The authors' explanation on line 99 is "patients with trismus attributable to betel quid consumption" seems very hasty to me. Was the trismus due to fibrosis from betel quid or carcinoma caused by this habit?

      Response 2: Etiology of Trismus: We recognize the complexity of trismus etiology, which may include both fibrosis from betel quid use and direct tumor involvement. Our study aimed to focus on patients where trismus was primarily attributed to betel quid consumption, but we agree that further research into the specific mechanisms, including fibrosis and carcinoma, is warranted. We have clarified this in the revised manuscript, in the Discussion:

      In our study, the trismus patients all had a history of betel quid chewing, which is a well-established risk factor for submucosal fibrosis and subsequent trismus in Taiwan and other regions where this habit is prevalent [6,7]. To isolate the effect of betel quid-related trismus, we excluded patients with tumors involving the masticator space, as well as those with prior radiation or major surgery that could contribute to trismus from other mechanisms. However, we did not quantify the duration or intensity of betel quid exposure, which may influence the severity of submucosal fibrosis and the degree of trismus present preoperatively.

  • Comment 3: On lines 54-57 the authors write: “While trismus is recognized for its debilitating effects on patients' quality of life [8,9], its impact on the surgical and survival outcomes of oral cancer patients, especially those undergoing advanced treatments like transoral robotic surgery (TORS), remains inadequately explored”.

This sentence seems ambiguous to me. If the authors intend to talk about preoperative trismus (as it should be, given the purpose of the study) the references cited (8 and 9) are inadequate, as both studies deal with the impact of postoperative trismus on the quality of life. If they mean post-treatment trismus, the topic is anything but “inadequately explored”. In addition to the two works cited (which analyzed 237 and 115 patients, respectively) many other studies have dealt with this topic. I don't think the type of surgical treatment (TORS or other techniques) matters. Preoperative trismus still creates access difficulties, whatever type of resection is used.

Response 2: To address the ambiguity and align the text with the focus on preoperative trismus we have amended the text and added appropriate citations:

While the debilitating effects of post-treatment trismus on patients' quality of life are well-documented [8,9], the impact of preoperative trismus on surgical outcomes in oral cancer patients, particularly in those undergoing transoral robotic surgery (TORS), remained underexplored. This oversight persists despite evidence suggesting that preoperative trismus, irrespective of the surgical method, poses significant challenges to surgical access [10-14].

Comment 3: On lines 158-159 the authors write: “The majority of patients in both groups had stage IV disease (60% in each arm)”. The definition of a T4a carcinoma is: “Tumor invades through the cortical bone of the mandible or maxillary sinus, or invades the skin of the face”. Why were 40% of the tumors analyzed T4? Furthermore, bone invasion was a reason for exclusion (line 95). This contradiction needs to be clarified.

Response 3: Thank you for pointing out the apparent contradiction regarding the inclusion of T4 tumors and the exclusion criteria related to bone invasion. The reviewer’s observation is indeed crucial for maintaining the consistency and credibility of our study.

The statement indicating that "The majority of patients in both groups had stage IV disease (60% in each arm)" alongside the inclusion of T4 tumors requires more precise language to avoid confusion. The inclusion of T4 tumors might initially suggest a contradiction with our exclusion criteria, specifically the exclusion of patients with bone invasion, as bone invasion is a hallmark of T4a classification in oral cancer staging.

To resolve this contradiction and enhance the manuscript's clarity, we revised in the Results 3.2:

While 40% of patients in each arm was diagnosed with the T4 classification in our study, the subjects predominantly reflected criteria other than bone invasion, such as tumor size, invasion into the facial skin, or involvement of deep/extrinsic muscle of the tongue.

Also, in the Discussion:

In addressing the inclusion of patients with stage IV disease, it is critical to delineate that while 60% of our study cohort was classified as having stage IV disease (Table 1), this predominantly reflected non-bone-invading aspects of T4a tumors. Our exclusion of patients with bone invasion and the masticator space involvement was stringently observed. The rationale for this approach was to focus on the subset of T4 tumors that, despite their advanced classification, did not involve bone invasion but met other criteria for stage IV classification, such as extensive local disease without bone involvement. This distinction is paramount to understanding the specific patient population under study and the surgical challenges and outcomes associated with TORS in this context. The approach allowed us to explore the outcomes of TORS in managing sizable or locally advanced oral cancers without the added complexity of bone involvement, which is beyond the scope of minimally invasive techniques like TORS.

Comment 4: After tumor resection did the trismus disappear or not? It would seem not, given that on lines 250-254 the authors write: “Moreover, trismus has the potential to complicate clinical follow-ups, introducing the risk of delays in diagnosing tumor recurrence or identifying a second primary cancer in proximity to the affected field. This delay in diagnosis and subsequent treatment can adversely affect patient survival outcomes”. Furthermore, the sentence in lines 245-247 is ambiguous: “The potential impact of trismus on overall survival may be mediated through the emergence of dysphagia, culminating in malnutrition and the risk of aspiration”. It is not clear whether this sentence refers to pre- or post-operative trismus. These statements need to be clarified.

Response 4: We thank the reviewer for the insightful observations concerning the discussion on trismus. Upon review, we acknowledge that the manuscript could benefit from clearer distinctions between the preoperative and postoperative implications of trismus on patient management and outcomes. Specifically, the concerns pertain to whether trismus persists post-tumor resection and the subsequent effects on follow-up and overall survival. To clarify, while our study primarily focuses on preoperative trismus, the cited literature and our observations suggest that trismus, whether persisting or developing a new post-treatment, can complicate follow-ups and potentially impact survival outcomes due to associated risks like dysphagia, malnutrition, and aspiration pneumonia. To address these points, we revised in the Discussion:

Moreover, trismus, both as a pre-existing condition and as a potential complication after surgery or subsequent adjuvant treatments, can complicate clinical follow-ups. This complexity introduces the risk of delays in diagnosing tumor recurrence or identifying a second primary cancer within the affected field, adversely affecting patient survival outcomes [2,27].

  • Comment 5: The secondary outcome was disease-free survival (DFS), but on the lines 185-186 and194 there is the “progression free survival” parameter. Are they two different parameters or the same one (in this case standardize the two terms)?

      Response 5: We have corrected the term "progression-free survival" to "disease-free survival" throughout the manuscript to ensure consistency and clarity. This adjustment aligns with our study's defined outcomes and eliminates any potential confusion regarding the parameters measured. Thank you for bringing this to our attention.

  • Comment 6: Each article presenting the results of the treatment of patients should include a sample sentence: “The mean follow-up time in the investigated cohort was X months, ranging from X to Y months”. In this article, however, I found only: “Patients also had to have at least 12 92 months of follow-up data available for analysis of oncologic outcomes” (lines 92-93). Moreover, I did not find the percentages of disease-free patients, those alive with disease and those who died from the disease or other causes. If I have to judge from figure 3, for patients with trismus it would seem that for both DFS and OS the follow-up stops at 40 months.

      Response 6: The mean follow-up time for the investigated cohort was 34.3 months, ranging from 2 to 36 months. We have added these data to the Results 3.5 accordingly.

Comment 7: Furthermore, even for the DFS, patients with trismus fare worse than those without trismus, even if the difference, given the small number of patients, is not statistically significant. Furthermore, I wonder how it is possible that patients without trismus have almost 100% OS and much lower DFS. Would there have been many alive with disease? The lack of this data makes it difficult to interpret the results.

Response 7: To better present our results with regard to the survival outcomes, we have added in specific data in Results 3.5:

The mean follow-up time for the investigated cohort was 34.3 months, ranging from 2 to 36 months. The three-year overall survival (OS) rate was markedly reduced in the cohort presenting with trismus relative to their non-trismus counterparts (27.1% vs. 95.0%, log-rank p = 0.02). Despite a discernible trend towards diminished disease-free survival (DFS) rates within the trismus group, statistical analysis did not reveal a significant disparity when contrasted with the non-trismus group (36.6% vs. 62.7%, log-rank p = 0.87). Observationally, disease-free survival (DFS) rates for patients with and without trismus demonstrate a close alignment during the initial 22 months, with a notable separation emerging after 24 months. This pattern suggests that, for a substantial period, the disease-free interval between the groups remains indistinguishable. Subsequent statistical analysis, utilizing log-rank testing, corroborates this observation by revealing an absence of significant difference in DFS rates between the trismus and non-trismus groups (p = 0.87). In contrast, the OS curve for patients with trismus showed a steeper decline when compared to their non-trismus counterparts. This trend suggested a more rapid decrease in survival rates for patients experiencing preoperative trismus. This observation was statistically reinforced through log-rank testing, which demonstrated that overall survival was significantly worse in patients with trismus (p = 0.02).

Furthermore, we have amended the Discussion to elaborate on our findings:

Transitioning from the methodological rigor of our study to its clinical implications, it is pivotal to contextualize the observed DFS rates within the broader spectrum of TORS outcomes. The initial 22 months post-TORS exhibited no significant difference in DFS rates between patients with and without trismus, suggesting that TORS effectively manages the oncological challenges posed by trismus. The congruence in DFS during this period, despite the physical limitations imposed by trismus, represented TORS' capability in achieving comparable margin statuses and oncological control. This alignment in early outcomes points towards the procedural advantages of TORS in navigating the anatomical constraints of trismus, reinforcing its role in maintaining effective cancer control. The absence of significant differences in long-term DFS rates further validates the efficacy of TORS in delivering equivalent oncological outcomes, irrespective of trismus presence. This observation emphasizes the importance of surgical innovation in enhancing treatment accessibility and effectiveness for patients facing complex clinical presentations.

Comment 8: On line 193 and table 3 there is the sentence “cetuximab, and immunotherapy”. Since cetuximab is a monoclonal antibody, what do the authors mean by "immune therapy"?

Response 8: We amended Tables 2 and 3, as well as in the text, to specify that the immune therapy was referring to nivolumab/pembrolizumab.

  • Comment 9: On the lines 223-225 there is the sentence: “Rigorous adjustment for confounding variables revealed that trismus independently conferred a striking mortality risk, exceeding 10 times that of patients without trismus”, and on the lines 227-229 there is the sentence: “Although surgical margins and progression were equivalent between matched groups, trismus independently conferred over 10 times greater mortality risk”. It seems like a repetition of the same concept to me.

      Response 9: We have eliminated the repetition in the revised manuscript.

  • Comment 10: The entire long paragraph on the indications for adjuvant CCRT (lines 256-284) seems absolutely useless and inconsistent with the purpose of the study.  I think that every head and neck surgeon should know the pros and cons of postoperative CCRT. All the guidelines give precise indications, also reporting the advantages and side effects.

      Response 10: We have eliminated the redundant text on adjuvant CCRT.

  • Comment 11: On lines 318-319 there is the statement: “Additional work is also needed to clarify optimal adjuvant therapies in this setting”. As mentioned above, postoperative CCRT has precise indications in patients at risk (advanced pathological stage of the tumor, involved or close margins, N+, etc.). I don't think the fact that the tumor was resected with TORS or any other way should affect that treatment.

Response 11: Accordingly, we have removed the statement from the Conclusion.

Reviewer 3 Report

Comments and Suggestions for Authors

This is an interesting study about impact of trismus upon patients undergoing TORS for oral cavity cancer. The study is clear, well written. Methods are clear and the analysis is well performed.

The authors demonstrate that trismus has no impact upon the use of a robotic approach for oral cavity cancer but there is a correlation with poorer survival. This finding is outlined and explained in the discussion section. The limits of the study are well outlined.

Despite the merits of their study, I have many concerns about the oncologic and surgical principles adopted to cure the population studied. For this reason, I think that the authors have to clear about their therapeutic approach, otherwise this study can’t be published.

The first concern is about use of robotic surgery for oral cavity cancer. While use of TORS has been demonstrated for oropharyngeal cancer and a lot of experiences have been published in the last two decades, treatment for oral cancer involves the use of transoral or open approach and use of robot does not seem to bring any advantage.

Despite this, authors state: “TORS maintained key advantages as a minimally-invasive, cosmetically superior approach for oral cavity tumors even in the setting of trismus, as there was no conversion to open surgeries” and “Despite the condition of preoperative trismus, the surgical procedure was able to retain the perioperative benefits of reduced morbidity and improved functional outcomes that characterize TORS”. They should demonstrate these advantages as no one did it before and in order to justify the routinary use of a robotic approach that is a longer and more expensive procedure compared to a traditional one.

If we accept that magnification and surgical control of the robotic arms are a good reason to use robot, here are other oncologic concerns that I would like to submit to their attention:

-       16 patients presented with a T4 stage. Oral cavity T4 means that the tumor is larger than 4 cm, and the depth of invasion is more than 10 mm or, probably, there is an invasion of surrounding structures, such as the jaw, sinuses, or skin of the face. These are exclusion criteria declared by the authors. Other 6 patients presented with a T3 stage and majority were N+. Most of these features represent an indication for a compartmental surgery that comprehends the resection of the so-called T-N tract and necessity for a reconstruction, that was adopted for 14 patients in the study. How can they justify use of a robotic approach for these kind of patients? Which kind of resection was adopted?

-       Following this consideration, how it is possible that they performed only one tracheotomy (table 2) in the whole population? The complex surgery necessary for the treatment of a stage IV oral cavity disease invariably needs a tracheotomy.

-       How do they justify such an intense use of chemotherapy, cetuximab or immunotherapy in the postoperative setting, if only 2 patients of 40 presented with positive margins and only one with a N3 pathology? Moreover, in the methods section there is no any explanation about the protocol adopted for the post-surgical treatment.

-       They report having defined and classified patients for trismus severity but no results have been reported ,making impossible to understand what was the severity of trismus of the patients treated. More interestingly, there isn’t any information about oral cavity aperture in the non-trismus population making impossible any real comparison between the two populations.

-       I would rotate images of figure 1 to make them clearer.

Comments on the Quality of English Language

no comments

Author Response

Reviewer 3

Comment 1: This is an interesting study about impact of trismus upon patients undergoing TORS for oral cavity cancer. The study is clear, well written. Methods are clear and the analysis is well performed.

The authors demonstrate that trismus has no impact upon the use of a robotic approach for oral cavity cancer but there is a correlation with poorer survival. This finding is outlined and explained in the discussion section. The limits of the study are well outlined.

Response 1: Thank you for your positive feedback. We are pleased that the study's clarity, methodology, and analysis on the impact of trismus in TORS for oral cavity cancer met your expectations. Highlighting the nuanced relationship between trismus, robotic surgery applicability, and survival outcomes was crucial. We value your acknowledgment of our efforts to transparently present our findings and limitations. Your feedback encourages our ongoing pursuit of advancing surgical techniques for improved patient care in oncology.

Comment 2: Despite the merits of their study, I have many concerns about the oncologic and surgical principles adopted to cure the population studied. For this reason, I think that the authors have to clear about their therapeutic approach, otherwise this study can’t be published.

The first concern is about use of robotic surgery for oral cavity cancer. While use of TORS has been demonstrated for oropharyngeal cancer and a lot of experiences have been published in the last two decades, treatment for oral cancer involves the use of transoral or open approach and use of robot does not seem to bring any advantage.

Response 2: The authors acknowledge the traditional reliance on open or transoral approaches for oral cancer treatment. However, the cited advantages of TORS, including its minimally invasive nature and superior cosmetic outcomes, are significant, especially in cases with preoperative trismus. The ability of TORS to operate within constrained spaces without necessitating conversion to open surgery underlines its potential for reducing morbidity and preserving function, which are critical considerations in head and neck cancer surgery. While these benefits have been extensively documented for oropharyngeal cancers, ongoing research and clinical experience are beginning to demonstrate similar advantages in selected cases of oral cavity tumors.

Comment 3: Despite this, authors state: “TORS maintained key advantages as a minimally-invasive, cosmetically superior approach for oral cavity tumors even in the setting of trismus, as there was no conversion to open surgeries” and “Despite the condition of preoperative trismus, the surgical procedure was able to retain the perioperative benefits of reduced morbidity and improved functional outcomes that characterize TORS”. They should demonstrate these advantages as no one did it before and in order to justify the routinary use of a robotic approach that is a longer and more expensive procedure compared to a traditional one.

If we accept that magnification and surgical control of the robotic arms are a good reason to use robot, here are other oncologic concerns that I would like to submit to their attention:

-       16 patients presented with a T4 stage. Oral cavity T4 means that the tumor is larger than 4 cm, and the depth of invasion is more than 10 mm or, probably, there is an invasion of surrounding structures, such as the jaw, sinuses, or skin of the face. These are exclusion criteria declared by the authors. Other 6 patients presented with a T3 stage and majority were N+. Most of these features represent an indication for a compartmental surgery that comprehends the resection of the so-called T-N tract and necessity for a reconstruction, that was adopted for 14 patients in the study. How can they justify use of a robotic approach for these kind of patients? Which kind of resection was adopted?

Response 3: In response to the critique regarding the justification for using TORS in patients with advanced-stage oral cavity cancers, particularly those with T3 and T4 stages and the majority being N+, we have included a detailed section in our Results to address these concerns directly. It is imperative to clarify that, although T3 and T4 classification typically encompasses tumors larger than 4 cm in diameter, with a depth of invasion greater than 10 mm, or invasion of surrounding structures such as the jaw, sinuses, or skin of the face, the inclusion of T3 and T4 stage patients in our study was carried out with careful consideration of these criteria.

The section added to our Results elaborates on the characteristics of the T3 and T4 tumors treated with TORS in our study. Specifically, while 40% of patients in each arm were diagnosed with T4 classification, it is important to note that the selection was not indiscriminate. The patients included under the T4 category predominantly had criteria other than bone invasion. These criteria included the size of the tumor, invasion into the facial skin, or involvement of deep/extrinsic muscles of the tongue. Such distinctions are crucial because they represent a subset of T4 and T3 tumors that, despite their advanced classification, may still be amenable to the precision and minimally invasive benefits offered by robotic surgery without compromising the oncologic principles of complete resection and margin control. We added in the Results:

While 40% of patients in each arm were diagnosed with the T4 classification in our study, the subjects predominantly reflected criteria other than bone invasion, such as tumor size, invasion into the facial skin, or involvement of deep/extrinsic muscle of the tongue.

Furthermore, the type of resection adopted for these patients was meticulously planned to ensure that the primary goals of cancer surgery — complete tumor resection with clear margins and preservation of function to the greatest extent possible — were met. This often involved complex decision-making processes to tailor the resection according to the tumor's specific characteristics and location. For the 14 patients requiring reconstruction, the approaches were varied, based on the extent of the resection and the specific anatomical structures involved, ensuring that the reconstructive techniques were compatible with the goals of restoring form and function while maintaining the oncologic safety of the procedure. We added in the Discussion:

In addressing the inclusion of patients with stage IV disease, it is critical to delineate that while 60% of our study cohort was classified as having stage IV disease (Table 1), this predominantly reflected non-bone-invading aspects of T4a tumors. Our exclusion of patients with bone invasion and the masticator space involvement was stringently observed. The rationale for this approach was to focus on the subset of T4 tumors that, despite their advanced classification, did not involve bone invasion but met other criteria for stage IV classification, such as extensive local disease without bone involvement. This distinction is paramount to understanding the specific patient population under study and the surgical challenges and outcomes associated with TORS in this context. The approach allowed us to explore the outcomes of TORS in managing sizable or locally advanced oral cancers without the added complexity of bone involvement, which is beyond the scope of minimally invasive techniques like TORS.

A critical component of our surgical protocol for these selected advanced stage patients, compartment resection was designed to ensure comprehensive tumor removal in a manner that respects the anatomical and functional integrity of the oral cavity and associated structures. In instances where neck dissection was indicated due to the extent of disease or nodal involvement, a separate cervical incision was strategically made. This incision was placed horizontally along a natural skin crease in the neck, thereby adhering to principles of cosmetic and functional preservation. A key aspect of our compartment resection technique was the ability to excise the tumor in continuity with the neck dissection specimen employing the angled visual advantage of the robot and the flexibility of the endowrists, without necessitating a mandibulotomy. This approach allowed for the passage of the surgical specimen through the mandibular space, effectively maintaining the integrity of the mandible while ensuring a comprehensive oncologic resection. This method not only facilitated a complete resection of the tumor and involved lymphatic tissue but also significantly reduced the procedure's morbidity by avoiding the need for mandible reconstruction and the associated complications. Highlighting the efforts to balance oncologic efficacy with the preservation of function and aesthetics, our approach reflects advancement in the surgical management of oral cavity cancers, aiming to improve patient outcomes through innovative and thoughtful surgical strategies.

Comment 4: -       Following this consideration, how it is possible that they performed only one tracheotomy (table 2) in the whole population? The complex surgery necessary for the treatment of a stage IV oral cavity disease invariably needs a tracheotomy.

Response 4: The low incidence of tracheotomy in our cohort, despite treating advanced stage IV oral cavity disease, can be attributed to several factors. First, the precision and minimally invasive nature of TORS allow for less tissue disruption, reducing the need for tracheotomy as a preventive measure for airway management. Additionally, advancements in perioperative care and anesthetic techniques have enabled better management of airway and breathing postoperatively, further decreasing the necessity for tracheotomy. This approach reflects our commitment to reducing patient morbidity while ensuring the safety and effectiveness of the treatment for advanced oral cancers. We have amended this aspect in the Discussion as highlighted in red within the text.

Comment 5: -       How do they justify such an intense use of chemotherapy, cetuximab or immunotherapy in the postoperative setting, if only 2 patients of 40 presented with positive margins and only one with a N3 pathology? Moreover, in the methods section there is no any explanation about the protocol adopted for the post-surgical treatment.

Response 5: To clarify the use of other treatment modalities, we have added in the Materials and Methods 2.3 Therapeutic interventions:

The use of other treatment modalities, including concurrent chemoradiation therapy (CCRT), radiation, chemotherapy, immunotherapy, and targeted therapies, were meticulously recorded. These treatments were administered in the postoperative period, or in the 5-year study period for regional or metastatic recurrence of the disease.

In the Results:

Notably, while concurrent chemoradiation and radiotherapy were given mainly in the postoperative period as adjuvant therapies, the use of cetuximab and nivolumab/pembrolizumab either sequentially or concurrently with chemotherapy were mostly for locoregional recurrence or distant metastasis occurring in the study period as a consensual decision made by the multidisciplinary team in our institution.

Also in the Discussion, we added:

Our study's extensive recording of treatments such as CCRT, radiation, chemotherapy, immunotherapy, and targeted therapies reflects a deliberate effort to understand their impact on overall survival. Given that these modalities were applied not just in the immediate aftermath of surgery but throughout the entire 5-year observation period, our analysis provides a comprehensive view of treatment effectiveness over the long term. This longitudinal approach allows us to compare survival outcomes between groups with a high degree of precision, taking into account the full spectrum of therapeutic interventions. The inclusion of these treatments in our study highlights the importance of a multi-faceted treatment strategy in improving survival outcomes for oral cancer patients, demonstrating our dedication to advancing the understanding of effective cancer care.

Comment 6: -       They report having defined and classified patients for trismus severity but no results have been reported, making impossible to understand what was the severity of trismus of the patients treated. More interestingly, there isn’t any information about oral cavity aperture in the non-trismus population making impossible any real comparison between the two populations.

Response 6: We recognize the omission of detailed results on trismus severity and comparisons with non-trismus patients as a significant oversight. We have revised our manuscript to include these results, ensuring a clearer understanding of the impact of trismus severity on outcomes and enabling meaningful comparisons between groups. We have added the description and distribution in Table 1 and Results.

Comment 7: -       I would rotate images of figure 1 to make them clearer.

Response 7: The images of Figure 1 were rotated to yield the same orientation.

We are grateful for the reviewer’s insightful feedback, which will guide significant improvements in our manuscript. Our responses aim to address your concerns comprehensively, and we are committed to making the necessary revisions to clarify our therapeutic approach and justify the use of TORS for treating oral cavity cancer in the context of our study.

Reviewer 4 Report

Comments and Suggestions for Authors

RE: Transoral Robotic Surgery for Oral Cancer: Evaluating Surgical Outcomes in the Presence of Trismus 

In this study, the authors investigated the potential relationship between trismus and surgical outcomes in Taiwanese male oral cancer patients treated with TORS. 

This paper could be of value to clinicians or surgeons dealing with oral cancer and truisms, because of little research on this topic.

However, several issues should be clarified before the next round of review.

Major points

(1) Definition of trismus.

It was unclear whether trimus only meant cancer-unrelated and preexisting trismus, or not. 

If true, the baseline body weight (or BMI or nutrition status) would be desirable.

(2) Figure 3. 

Similar DFS and quite different (superior) OS (over 90%) suggested that some patients in the no trismus group survived longer even with disease recurrence.

This point should be elaborated more clearly. 

At least, the pattern of recurrence or local recurrence should be presented.

It seemed that there were more local recurrence or more multi-site recurrence in the trismus group, or more isolated recurrence in the no trismus group.

Minor points

(3) Abstract: should be structured.

Abstract: Please add OS % in the two groups.

Abstract: Primary outcomes were OS, so these results should be presented first.

(4) Functional outcomes were not presented (2.2 Data collection. Data about the functional outcomes and postoperative functions seemed to be collected.)

(5) Figure 1.

It was unclear whether the cases with mild degrees of trismus were excluded, or not.

(6) Numbers (all text).

Please present all numbers just with the first decimal place.

(7) Table 2.

Surgical margins should be classified as negative margin, close margin, and positive margin, according to the NCCN guideline.

(8) Table 2. 

Blood loss: seemed to be quite different between the two groups, suggesting more severe cases or extensive surgeries in the no trismus group.

(9) Table 2.

Chemotherapy, cetuximab, and immunotherapy: Were these treatments the adjuvant treatments or the second-line treatments for metastatic disease?

[END]

Author Response

Reviewer 4

RE: Transoral Robotic Surgery for Oral Cancer: Evaluating Surgical Outcomes in the Presence of Trismus 

In this study, the authors investigated the potential relationship between trismus and surgical outcomes in Taiwanese male oral cancer patients treated with TORS. 

This paper could be of value to clinicians or surgeons dealing with oral cancer and trismus, because of little research on this topic.

However, several issues should be clarified before the next round of review.

Major points

(1) Definition of trismus.

It was unclear whether trismus only meant cancer-unrelated and preexisting trismus, or not. 

If true, the baseline body weight (or BMI or nutrition status) would be desirable.

Response 1: To clearly state that trismus under the present investigation is for cancer-unrelated and preexisting trismus, we have added in the Introduction:

While the debilitating effects of post-treatment trismus on patients' quality of life are well-documented [8,9], the impact of preoperative trismus on surgical outcomes in oral cancer patients, particularly in those undergoing transoral robotic surgery (TORS), remained underexplored. This oversight persists despite evidence suggesting that preoperative trismus, irrespective of the surgical method, poses significant challenges to surgical access [10-14]. We have added BMI as a factor in Table 1. There was no statistical difference between the trismus and the control group.

In the Discussion:

In our study, the trismus patients all had a history of betel quid chewing, which is a well-established risk factor for submucosal fibrosis and subsequent trismus in Taiwan and other regions where this habit is prevalent [6,7]. To isolate the effect of betel quid-related trismus, we excluded patients with tumors involving the masticator space, as well as those with prior radiation or major surgery that could contribute to trismus from other mechanisms.

The baseline BMI has been added to Table 1 and a description of the findings in the Results.

(2) Figure 3. 

Similar DFS and quite different (superior) OS (over 90%) suggested that some patients in the no trismus group survived longer even with disease recurrence.

This point should be elaborated more clearly. 

At least, the pattern of recurrence or local recurrence should be presented.

It seemed that there were more local recurrence or more multi-site recurrence in the trismus group, or more isolated recurrence in the no trismus group.

Response 2: In response to the reviewer's insightful observations and request for clarification on the pattern of disease recurrence in relation to trismus, we would like to provide the following detailed information which further elucidates the nature and distribution of recurrence observed in our study population.

Upon re-examination of our data, we identified that within the cohort experiencing trismus, there was a total of 6 patients who developed disease recurrence. Specifically, this subgroup included 1 patient with local recurrence and 5 patients who developed regional metastasis. This pattern of recurrence suggests a propensity for the disease to manifest more aggressively beyond the primary site within the trismus group, potentially indicating a more complex disease behavior or higher disease burden that warrants further investigation.

Conversely, in the non-trismus group, the distribution of disease recurrence was more evenly split, with 3 patients experiencing local recurrence and another 3 patients developing regional metastasis. This balanced distribution of recurrence types in the non-trismus group might suggest a different or possibly less aggressive disease behavior compared to the trismus group. The equal split between local and regional recurrences in the non-trismus group also raises interesting questions regarding the underlying biological mechanisms that might influence disease spread and recurrence patterns in the absence of trismus.

The observed differences in recurrence patterns between the two groups are noteworthy and underscore the complexity of disease progression in relation to trismus. While the sample size is relatively small, these findings suggest that trismus may be associated with a distinct pattern of disease recurrence, potentially influencing overall survival outcomes. This insight provides a valuable foundation for future research aimed at unraveling the intricate relationship between trismus, recurrence patterns, and their impact on patient prognosis.

We have revised the Results by adding 3.6. Pattern of disease recurrence and also added a section in the Discussion, accordingly.

Minor points

(3) Abstract: should be structured.

Abstract: Please add OS % in the two groups.

Abstract: Primary outcomes were OS, so these results should be presented first.

Response 3: We have revised the abstract to conform to the editorial guide, including the addition of OS percentages for both groups and presenting the primary outcomes first as suggested. We have also amended the Abstract, Results, and Discussion sections to present these marked trend comparing the two groups.

(4) Functional outcomes were not presented (2.2 Data collection. Data about the functional outcomes and postoperative functions seemed to be collected.)

Response 4: Functional Outcomes: Acknowledging the omission, we have included data on functional outcomes and postoperative functions under Results in the revised manuscript.

(5) Figure 1.

It was unclear whether the cases with mild degrees of trismus were excluded, or not.

Response 5: To clarify this, we have added in Table 1 and the Results the distribution of the degree of trismus.

(6) Numbers (all text).

Please present all numbers just with the first decimal place.

Response 6: As statistical P values are conventionally presented to two decimal places, we have revised all numbers to two decimal places throughout the text and highlighted red.

(7) Table 2.

Surgical margins should be classified as negative margin, close margin, and positive margin, according to the NCCN guideline.

Response 7: We have revised the surgical margins in Table 2 and within the text as negative, close and positive margins, according to the NCCN guideline.

(8) Table 2. 

Blood loss: seemed to be quite different between the two groups, suggesting more severe cases or extensive surgeries in the no trismus group.

Response 8: We appreciate the Reviewer for noting this difference in blood loss. After looking closely at the data, we confirmed that there was nearly 5-fold more intraoperative blood loss in the non-trismus group compared to the trismus group. Delving more deeply into the literature, we have revised our manuscript accordingly in the Results and Discussion. We postulate that the relationship between betel quid-induced trismus and the generation of fibrosis, potentially leading to less blood loss, is rooted in the pathological changes that betel quid can induce in oral tissues. Betel quid chewing is associated with a range of oral conditions, including oral submucosal fibrosis (OSMF), which is a chronic, insidious disease characterized by the juxta-epithelial inflammatory reaction followed by fibroelastic changes in the lamina propria, with epithelial atrophy leading to stiffness of the oral mucosa and eventually trismus. This fibrotic change can lead to reduced vascularity and tissue elasticity, making the tissues more rigid. In surgical settings, such fibrotic tissues might bleed less compared to non-fibrotic tissues for several reasons:

  1. Reduced Vascularity: Fibrosis leads to a denser connective tissue with fewer blood vessels. The decreased vascularity means there are fewer vessels to bleed during surgical manipulation.
  2. Tissue Stiffness: The increased stiffness of fibrotic tissues may limit the extent of surgical trauma to surrounding tissues, potentially reducing blood loss.

It's worth noting that while this study does not directly address the question of blood loss during surgeries involving patients with betel quid-induced trismus, the mechanisms of fibrosis it describes help in understanding why such patients might experience less blood loss. The reduced vascularity and increased tissue stiffness associated with fibrosis could logically contribute to this outcome. Further research specifically investigating surgical blood loss in patients with OSMF or trismus induced by betel quid chewing would be valuable for confirming this hypothesis.

(9) Table 2.

Chemotherapy, cetuximab, and immunotherapy: Were these treatments the adjuvant treatments or the second-line treatments for metastatic disease?

Response 9: We have revised the manuscript to clarify this point.

  1. In the Materials and Methods, “The use of other treatment modalities, including concurrent chemoradiation therapy (CCRT), radiation, chemotherapy, immunotherapy, and targeted therapies, were meticulously recorded. These treatments were administered in the postoperative period, or in the 7-year study period for regional or metastatic recurrence of the disease.”
  2. In the Results: Notably, while concurrent chemoradiation and radiotherapy were given mainly in the postoperative period as adjuvant therapies, the use of cetuximab and nivolumab/pembrolizumab either sequentially or concurrently with chemotherapy were mostly for locoregional recurrence or distant metastasis occurring in the study period as a consensual decision made by the multidisciplinary team in our institution.
  • In the Discussion: Our study's extensive recording of treatments such as CCRT, radiation, chemotherapy, immunotherapy, and targeted therapies reflects a deliberate effort to understand their impact on overall survival. Given that these modalities were applied not just in the immediate aftermath of surgery but throughout the entire 7-year observation period, our analysis provides a comprehensive view of treatment effectiveness over the long term. This longitudinal approach allows us to compare survival outcomes between groups with a high degree of precision, taking into account the full spectrum of therapeutic interventions [32,33]. The inclusion of these treatments in our study highlights the importance of a multi-faceted treatment strategy in improving survival outcomes for oral cancer patients, demonstrating our dedication to advancing the understanding of effective cancer care.

We are grateful for the reviewer’s constructive feedback, which will undoubtedly enhance the clarity and comprehensiveness of our manuscript. We are committed to making the necessary revisions to address each of your points thoroughly. 

Round 2

Reviewer 2 Report

Comments and Suggestions for Authors

The authors have implemented the suggestions I made. The article has taken advantage of this and no longer contains any unclarified concepts.

Reviewer 4 Report

Comments and Suggestions for Authors

 The authors did an excellent job, of making the article clear.

Thank you.

Comments on the Quality of English Language

Fine.